# Prophylactic Administration of Perampanel for Post-Stroke Epilepsy (PROPELLER Study): A Trial Protocol

**DOI:** 10.3390/mps7050079

**Published:** 2024-10-05

**Authors:** Shuichi Yamada, Ichiro Nakagawa, Masashi Kotsugi, Kiyoshi Asada, Masato Kasahara

**Affiliations:** 1Department of Neurosurgery, Nara Medical University, 840 Shijo-cho, Kashihara 634-8522, Nara, Japan; nakagawa@naramed-u.ac.jp (I.N.); igustok@naramed-u.ac.jp (M.K.); 2Institute for Clinical and Translational Science, Nara Medical University, 840 Shijo-cho, Kashihara 634-8522, Nara, Japan; kasada@naramed-u.ac.jp (K.A.); kasa@naramed-u.ac.jp (M.K.)

**Keywords:** post-stroke epilepsy, prophylactic treatment, perampanel, prospective interventional study

## Abstract

Background: Post-stroke epilepsy can reduce patients’ abilities to carry out various activities of daily living. Despite their importance in preventing the onset of post-stroke epilepsy, the prophylactic administration of antiepileptic drugs is controversial due to a lack of high-level clinical research. In this study, we initiated a prospective interventional study of prophylactic antiepileptic drug administration in patients with a subcortical hemorrhage, who are at the highest risk of developing epilepsy after experiencing a stroke. Methods: The study was conducted in a single-center setting and was a single-arm study with no control group; the case entry period started in November 2023 and is due to end in March 2025. Only cases with a subcortical hemorrhage will be included. The treatment regimen used in this study is 2 mg of perampanel per day. Perampanel will be administered for one year, followed by two years of follow-up, for a total study period of three years. The primary endpoint will be the development of epilepsy. Results: Perampanel administration is expected to reduce the incidence of post-stroke epilepsy in comparison to the results of previous reports on the use of alternative treatments. Conclusions: The results of this study will provide new insights into the prevention of post-stroke epilepsy. The relatively small size of this study makes it difficult to provide strong evidence of the efficacy of perampanel, but it may serve as a basis for larger clinical trials.

## 1. Introduction

A stroke may consist of cerebral infarction, a cerebral hemorrhage, or a subarachnoid hemorrhage; patients may subsequently develop new-onset epilepsy, which is also referred to as post-stroke epilepsy. This is reported to occur in 5–10% of stroke survivors. The number of new post-stroke epilepsy cases in Japan is estimated to be approximately 10,000 per year, so it is by no means a rare disease [1].

Post-stroke epilepsy can cause a decrease in the ability of patients who have survived the acute stage of a stroke—that is, stroke survivors—to carry out various activities of daily living (ADLs), and so preventing and treating post-stroke epilepsy are particularly important. There have been many reports on the risk factors associated with developing post-stroke epilepsy [2,3,4], but there are only a few reports on potential drug therapies that can prevent post-stroke epilepsy [5,6]. The Japanese guidelines mention the prophylactic administration of antiepileptic drugs for high-risk cases [7], but they only state that “administration may be considered,” and both the number of recommendations for this treatment modality and evidence of its efficacy are limited.

Furthermore, there are no drugs approved for prophylactic use, and doses and administration periods have also yet to be optimized. Although the American Stroke Association (ASA) guidelines mention the prophylactic administration of antiepileptic drugs for the treatment of a subarachnoid hemorrhage [8], there are few recommendations for this use and there is no mention of specific drugs or administration periods. According to the ASA, the prophylactic administration of drugs is currently being carried out across the world, but this prevalence is unknown.

The aim of this study is to demonstrate the effectiveness of the prophylactic administration of antiepilepsy drugs for the treatment of post-stroke epilepsy. In this study, the drug under investigation is perampanel. This drug was chosen due to the fact that it is regarded as the most analogous pharmaceutical agent to the pharmacological mechanism of post-stroke epilepsy. The specifics of the selected drug are outlined in Section 3.2.

The results of this study will provide new insights into the prevention of post-stroke epilepsy and will significantly contribute to preventing a decline in the ability of stroke survivors to perform ADLs. This study’s organization is shown in jRCTs051230107 on 4 October 2023.

## 2. Experimental Design

### 2.1. Study Design and Setting

The aim of this study is to evaluate the usage of perampanel for the prevention of post-stroke epilepsy in patients with a subcortical hemorrhage in a single-center, single-arm study. The rationale for conducting a single-arm study without a placebo group is that we anticipate it to be challenging to recruit the requisite number of patients at the study sites. Further details are provided in Section 2.4. All participants will be recruited from the Department of Neurosurgery at Nara Medical University Hospital (Kashihara, Japan).

### 2.2. Eligibility Criteria

The target stroke type is limited to subcortical hemorrhages. We believe that this will allow us to observe the effectiveness of prophylactic administration. Within the existing literature, including in a previous study by our group, it has been reported that a subcortical hemorrhage is the most common cause of post-stroke epilepsy [2,3,4,9]. A diagnosis of a subcortical hemorrhage is made through the use of computed tomography (CT) or magnetic resonance (MR) imaging of the head during the patient’s initial admission to hospital. A cerebral subcortical hemorrhage is defined as a hemorrhage involving the cortex of the frontal, temporal, parietal, and occipital lobes. The CAVE score is a well-established risk assessment measure that is used to predict the onset of post-stroke epilepsy in patients with a cerebral hemorrhage [10]. However, the CAVE score includes the occurrence of early seizures, and patients who experience an early seizure should receive antiepileptic drugs immediately, rendering these patients unsuitable for inclusion in this study. Consequently, the CAVE score will not be utilized in the selection of patients for this study.

Patients with a subcortical hemorrhage who are available for registration in the study within 1 week, whose age at the onset is between 50 and 90 years, who are capable of taking drugs orally, including via tube feeding, who have been fully informed of the details of this clinical research, and who have given their written consent, either themselves or through a surrogate, are eligible for inclusion in this study.

Exclusion criteria include subjects with a history of epilepsy; patients currently taking antiepileptic drugs; patients whose cerebral hemorrhage is known to be caused by a vascular malformation such as arteriovenous malformation or Moyamoya disease (CT angiography, MR angiography, or digital subtraction angiography can be utilized for the diagnosis of vascular malformations) or is traumatic in nature; patients with a severe renal or hepatic impairment; patients who are pregnant or breast feeding; patients with an allergy to the target drug; and those judged to be inappropriate as research subjects by the principal investigator.

### 2.3. Discontinuation of Interventions

The investigator will discontinue the study if any of the following criteria are met: the onset of epilepsy, worsening complications, the development of a disease which is suspected of being as a result of this research, pregnancy, an allergy to the test drug, severe renal or hepatic dysfunction, difficulty in continuing to take the drug, and/or the research subject is deemed ineligible to continue oral administration for other reasons. The investigator will withdraw patients from the study if any of the following conditions are met: a research subject requests to be removed from the study or withdraws consent, a research subject is not eligible after registration, the entire study is canceled, and/or the doctor deems it appropriate to cancel the study for other reasons.

### 2.4. Sample Size

About 40 patients with a subcortical hemorrhage are admitted to the Nara Medical University hospital per year. During each registration period, which lasts for one year and six months, 54 patients (90% of 60 patients) are expected to be registered. Taking a drop-out ratio of 10% into account, approximately 48 patients will be eligible for analysis during the 3-year observation period.

Based on previous research, we know that the rate of developing epilepsy within three years after a stroke is approximately 20% [9,10,11]. If the incidence of post-stroke epilepsy can be suppressed to less than 10%, or to a maximum of 4 out of 48 cases, by the prophylactic administration of perampanel, the upper limit of the 95% confidence interval will be less than 20%, suggesting a certain level of effectiveness.

### 2.5. Data Collection and Management

The results will be collected by the investigators or outcome assessors using a case report form (CRF). The investigator will evaluate patients’ eligibility using routine blood and image examinations. An investigator will ask the participants or proxies about their medication status, the occurrence of epileptic seizures or a stroke, their survival status, and the occurrence of other adverse events every three months by telephone after the initiation of the treatment protocol. Epileptic seizures are defined as the acute onset of tonic or clonic motor convulsions, absence seizures leading to an attenuation of consciousness or a loss of consciousness, automatism, sensory abnormalities, and autonomic neurological symptoms. The investigator will determine whether the symptoms are indicative of epileptic seizures based on the information provided by the patient or a witness. The diagnosis of post-stroke epilepsy is based on the occurrence of a single seizure. Typically, epilepsy is diagnosed based on the occurrence of two or more seizures that occur at a certain interval. However, as reported by Fisher et al., post-stroke epilepsy can be diagnosed based on a single seizure due to its high recurrence rate [12]. This study also adopted this diagnostic criterion.

All information in the study protocol will be collected by the investigator during the study period and recorded on CRFs that will be stored at the Nara Medical University data center.

### 2.6. Statistical Methods

The number and proportion of epilepsy cases three years after the initiation of treatment will be counted, and the 95% confidence interval of the incidence rate will be calculated using the Clopper–Pearson method. The subjects used to calculate these proportions are those in the analyzed population who have survived for up to three years following the initiation of treatment who are available for follow-up, or those who have developed epilepsy within three years of initiating treatment.

In addition, as a secondary analysis, a Kaplan–Meier curve will be drawn for the number of days until the onset of epilepsy starting from the beginning of the treatment protocol, and the incidence rate of epilepsy within the three-year study period and its 95% confidence interval will be calculated using Greenwood’s calculation formula. At that time, patients who are unable to be followed-up with or who have died will be excluded from the analysis. We will also perform a secondary analysis with mortality as a competing risk.

Regarding the above two analyses, we will also conduct separate analyses for the Glasgow Coma Scale (GCS) and the National Institutes of Health Stroke Scale (NIHSS).

### 2.7. Monitoring

To improve the accuracy of data collection, a committee of staff who are not involved in this study will monitor the data gathered from the first patient at specified time points. The committee will deliver the results to the principal investigator. No interim analyses have been planned. We define adverse events as detrimental effects on a patient’s health due to the administration of a drug. The investigators will report to the Data and Safety Monitoring Board (DSMB) and the Minister of Health, Labour, and Welfare within the established submission period if a serious adverse event occurs. We did not establish a coordinating center or trial steering committee because this trial will be conducted at a single center. We organized the DSMB according to Japan’s Clinical Trials Act (Act No. 16 of 14 April 2017). The DSMB’s members are independent of the investigators and have experience in evaluating clinical trials. The DSMB will deliberate on the directions and continuation of this trial if unpredicted serious adverse events occur. This study will not be audited. However, an audit will be considered if the monitoring reveals serious violations of relevant laws and regulations or deviations from the protocol.

## 3. Procedure

### 3.1. Recruitment

If the team providing care considers a patient eligible for this study, they can refer the patient to the principal researcher. Written informed consent will be required from all patients or representative consenters prior to their inclusion in the study. Each participant will be informed that their participation is voluntary and that they will be able to withdraw from the study at any time.

### 3.2. Intervention Description

The antiepileptic drug that we chose to use is perampanel. The reasons for this decision are as follows: The mechanism behind the onset of post-stroke epilepsy is an increase in glutamate in the synaptic cleft that is caused by cell damage. In this case, the inhibition of glutamate release from the presynaptic membrane, which is the mechanism of action of many antiepileptic drugs, is not effective. In contrast, the mechanism of action of perampanel is the blockade of α-amino-3-hydroxy-5-methyl-4-isoxazolepropionic acid (AMPA) receptors, which are glutamate receptors located in the synaptic dura mater, which is consistent with the pathogenesis of post-stroke epilepsy. The prophylactic administration of levetiracetam, one of the most commonly utilized antiepileptic drugs globally, has also been documented for the treatment of post-stroke epilepsy [13,14]. In this study, we concentrated on the mechanism behind the onset of post-stroke epilepsy and utilized perampanel, which is regarded as more efficacious, with the objective of demonstrating the enhanced efficacy of this treatment protocol.

The treatment protocol used in this study is the off-label prophylactic administration of 2 mg of perampanel orally once daily before bedtime (Figure 1). Oral administration will be initiated within 1 week of the onset of the cerebral hemorrhage and the duration of the administration of the drug should be 1 year. This is because many guidelines state that long-term prophylactic administration is undesirable. During the drug administration period, in addition to the previous items, adherence to the medication will be checked every 3 months. Subjects will only continue to participate in the study if their adherence rate is at least 80%.

### 3.3. Participant Timeline

All outcomes will be evaluated, and data will be collected according to the participant’s timeline (Table 1). Neurological findings, blood tests, and imaging studies will be performed when a patient enters the study to confirm whether or not the patient is eligible to participate in the study. After the completion of the drug administration protocol, the patients will be followed-up with for another 2 years. Patient status should be checked every 90 days (allowed within ± 10 days) during the administration and follow-up periods and it will be confirmed through a telephone interview or outpatient visit. During the follow-up, patients will be checked for survival, the occurrence of seizures, the occurrence of another stroke, and adverse events.

### 3.4. Outcomes

The primary endpoint will be the percentage of patients who develop epilepsy within the first three years after the initiation of the treatment protocol. According to the International League Against Epilepsy, the definition of epilepsy, with a high likelihood of a persistently lowered seizure threshold and therefore a high risk of recurrence of events such as a stroke, is the occurrence of at least one epileptic seizure [10]. The secondary endpoints will be all-cause mortality within three years after the initiation of the treatment protocol and the occurrence of another stroke (cerebral infarction, cerebral hemorrhage, subarachnoid hemorrhage) within three years of starting the treatment protocol (regardless of whether this is a new stroke or a recurrence).

### 3.5. Ethics and Dissemination

#### 3.5.1. Ethics Approval and Consent to Participate

The Nara Medical University Certified Review Board (Approval Number: CRB5200002) approved the study protocol. Written informed consent will be obtained from all patients before their inclusion in the study, and each patient will be informed that their participation is entirely voluntary and that they may withdraw from the study at any time.

#### 3.5.2. Competing Interests

The authors declare that they have no competing interests.

Eisai (Eisai Co., Ltd. Tokyo, Japan), which produces and distributes perampanel in Japan, provided our hospital with nonfinancial support.

#### 3.5.3. Protocol Amendments

If the protocol requires changing, all revisions and their rationales will be reported to the approved Clinical Research Review Committee for review and approval. The principal investigator will then review the explanatory records of the patients.

#### 3.5.4. Confidentiality

All collected data will be stored anonymously on a computer that is not connected to the Internet at Nara Medical University. Only the investigators will be able to access the data. Appointed staff will be able to temporarily access the data during monitoring.

#### 3.5.5. Access to Data

The datasets used and/or analyzed during the study will not be available to the public but will be available from the corresponding author upon reasonable request.

#### 3.5.6. Dissemination Policy

The results of this research will be published in peer-reviewed journals, presented at national and international conferences, and distributed to participating physicians and participants.

## 4. Expected Results

The expected results of our study include a significantly lower incidence of post-stroke epilepsy in patients treated with perampanel than previously reported (5 to 10%). It may be feasible to ascertain certain risk factors for the emergence of post-stroke epilepsy based on data gathered at the time of admission. In particular, patients with high scores on the GCS and NIHSS are more likely to have extensive brain lesions and may be more susceptible to developing post-stroke epilepsy.

If the number of patients included in the study is small, there may be no cases of post-stroke epilepsy. In this case, it may be difficult to perform statistical analyses. However, these results may facilitate the design and planning of larger prospective studies, and this study may serve as a basis for subsequent studies that can provide more solid evidence. Although the existence of post-stroke epilepsy has been known for some time, there is very little evidence to support the prophylactic use of antiepileptic drugs to treat this condition. This may be due to the difficulty of conducting high-level prospective clinical trials. The complexity of the different pathological states of stroke patients, the complexity of the prognosis of individual patients, and the fact that, unlike other diseases, these patients are often transferred to other hospitals or rehabilitation centers instead of being discharged because of their reduced ability to perform ADLs as a result of the stroke, which makes follow-up difficult, may contribute to the difficulty in conducting this type of research. The results of this relatively small study are expected to be very significant. We hope that the success of this study will serve as a basis for new research in the future, and it is our hope that the aggregation of these studies will yield compelling evidence.

## Figures and Tables

**Figure 1 mps-07-00079-f001:**
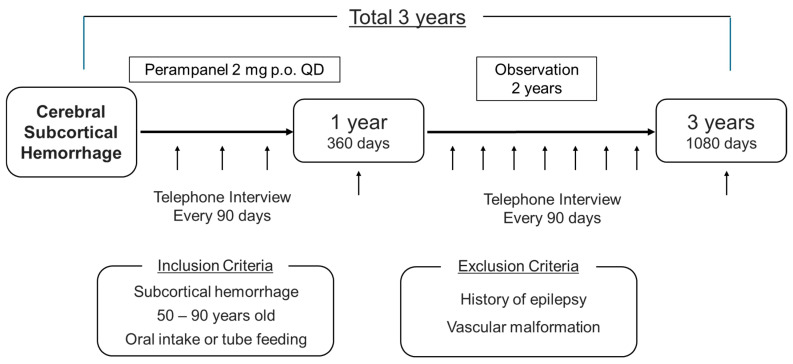
Schema of the study.

**Table 1 mps-07-00079-t001:** Schedule of the study.

	Entry	Intervention	Follow Up
Visit		1st90days	2nd180days	3rd270days	4th360days	5th450days	6th540days	7th630days	8th720days	9th810days	10th900days	11th990days	12th1080days
InformedConsent	○												
Patient’s Information	○												
PerampanelAdministration	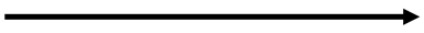						
Observation		○	○	○	○	○	○	○	○	○	○	○	○

## Data Availability

No new data were created or analyzed in this study. Data sharing is not applicable to this article.

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
