# Peer review of "Prophylactic Administration of Perampanel for Post-Stroke Epilepsy (PROPELLER Study): A Trial Protocol"

_mps, 2024, doi:10.3390/mps7050079_

Round 1

Reviewer 1 Report

Comments and Suggestions for Authors

This is the protocol of a prospective interventional single-arm uncontrolled single-center study of perampanel as prophylactic treatment of post-stroke epilepsy in patients with subcortical haemorrhage.

There are major flaws in the study design.

The lack of a control placebo group does not allow to draw any evidence about the study outcome.

An enrichment approach based on the CAVE score to select patients at the highest risk of post stroke epilepsy would increase the study power.

Why only subcortical haemorrhage will be included? What are the criteria used to define the subcortical haemorrhage?

Comments on the Quality of English Language

Improvement needed

Author Response

Thank you for your comments. I reply to them.

1.The lack of a control placebo group does not allow to draw any evidence about the study outcome.

We recognize that this is an important issue in this study and that, strictly speaking, it is preferable to compare the results to a placebo control.
However, in predicting the number of cases based on historical data, it was anticipated that it would be difficult to enroll a sufficient number of cases to analyze if a placebo group was established.
Therefore, priority was given to enrolling a solid number of cases only in the actual drug group without a placebo group. This has been noted in the text.

2. An enrichment approach based on the CAVE score to select patients at the highest risk of post stroke epilepsy would increase the study power.

Limiting the study to cases with high CAVE scores (e.g., 4 points) would significantly reduce the number of cases that could be included in the study, and the study may not be valid. Also, 4 points would indicate the onset of early seizures, and antiepileptic drugs are likely to be administered as treatment in such cases. Our protocol should be limited to patients who have never had a seizure. This has been noted in the text.

3. Why only subcortical haemorrhage will be included? What are the criteria used to define the subcortical haemorrhage?

We included subcortical hemorrhages in this study because previous studies have shown that the risk of post-stroke epilepsy is higher in subcortical hemorrhages than in other regions. Other studies have also shown an increased risk of post-stroke epilepsy in lesions involving the cortex and in hemorrhagic lesions.
Subcortical hemorrhage was defined as hemorrhage involving the cortex of the frontal, temporal, parietal, and occipital lobes. This has been added to the text.

Reviewer 2 Report

Comments and Suggestions for Authors

1) You should mention perampanel in the introduction. You only referred to it as an anti-epileptic drug without specifying which drug you used.

2It seems that you discussed the effect of perampanel on subarachnoid hemorrhage post-epilepsy, but in Figure 1, you referred to cerebral subarachnoid hemorrhage. Please clarify this and provide more information about the type of stroke you studied in the introduction.

3) The method of detecting SAH should be mentioned in the Methods section.

4)The epilepsy assessment should be mentioned in the Methods section

5) In the Expected “Results section”, you mentioned that "it may be difficult to perform statistical analyses". However, you explained using the Clopper-Pearson method and a Kaplan-Meier curve for statistical analysis. If analysis is difficult, why did you define these methods, and how did you determine that perampanel affects post-stroke epilepsy?

6) The data should be available, and the results should be organized into sections, addressing each graph/parameter you studied.

7) The Glasgow Coma Scale (GCS) and National Institutes of Health Stroke Scale (NIHSS) that you mentioned in the statistical analysis were not mentioned in the Expected Results section

8) You missed the Discussion section.

Author Response

Thank you for your comments. I reply to them.

1) You should mention perampanel in the introduction. You only referred to it as an anti-epileptic drug without specifying which drug you used.

We will describe about Perampanel in the introduction.

2) It seems that you discussed the effect of perampanel on subarachnoid hemorrhage post-epilepsy, but in Figure 1, you referred to cerebral subarachnoid hemorrhage. Please clarify this and provide more information about the type of stroke you studied in the introduction.

In this study, we are not looking at “cerebral subarachnoid hemorrhage” but rather “cerebral subcortical hemorrhage. This study does not address SAH.

3) The method of detecting SAH should be mentioned in the Methods section.

As I answered in 2).
I added to the text about the method of detecting subcortical hemorrhage because there was no description of how to detect it.

4)The epilepsy assessment should be mentioned in the Methods section

Added to “2.5 Data collection and management” in “2. Experimental Design”.

5) In the Expected “Results section”, you mentioned that "it may be difficult to perform statistical analyses". However, you explained using the Clopper-Pearson method and a Kaplan-Meier curve for statistical analysis. If analysis is difficult, why did you define these methods, and how did you determine that perampanel affects post-stroke epilepsy?

The statement "It may be difficult to perform statistical analyses" in "4 Expected Results" is hypothetical story. We are discussing the statistical methods that should be employed should the study progress in a manner consistent with our expectations. As indicated in "4 Expected Results," we will present the complete results of this study as a foundation for further research should the statistical analyses prove challenging to execute. We believe this has a certain significance and effect.

6) The data should be available, and the results should be organized into sections, addressing each graph/parameter you studied.

This is a protocol paper. The results are not yet available.

7) The Glasgow Coma Scale (GCS) and National Institutes of Health Stroke Scale (NIHSS) that you mentioned in the statistical analysis were not mentioned in the Expected Results section.

Patients with high GCS and NIHSS scores are likely to have extensive lesions and may develop post-stroke epilepsy.
We added a note to the manuscript about this.

8) You missed the Discussion section.

The journal "Protocol and Method" does not have a discussion section. We believe that "Expected results" section is a discussion at this point in time, when the results are not yet available.
We have enhanced the description of this section, including the above.

Reviewer 3 Report

Comments and Suggestions for Authors

This is a protocol paper that is investigating the impact of prophylactic administration of perampanel for post-stroke epilepsy. I think the authors have done a a good job of writing up the protocol. I think one minor revision is required. Both the figure and table have perampanel have underlined in red, can this be revised?

Additionally, I would like to see a discussion of other post-stroke epilepsy medications that have been used.

Comments on the Quality of English Language

No comments, English appears to be fine.

Author Response

Thank you for your comments. I reply to them.

1. Both the figure and table have perampanel have underlined in red, can this be revised?

We consider this an error before proofreading. We will correct it accordingly. 

2. Additionally, I would like to see a discussion of other post-stroke epilepsy medications that have been used.

We added a description of other antiepileptic drugs in the section describing the reasons for selecting perampanel (3. Procedure 3.2 Intervention description).

Round 2

Reviewer 1 Report

Comments and Suggestions for Authors

The Authors have partially addressed the limitations of the study protocol. This study design will limit the strength of the results and conclusions.

Comments on the Quality of English Language

None

Reviewer 2 Report

Comments and Suggestions for Authors

Thank you for addressing all concerns.